# Tirzepatide and Glycemic Control Metrics Using Continuous Glucose Monitoring in Older Patients with Type 2 Diabetes Mellitus: An Observational Pilot Study

**DOI:** 10.3390/geriatrics9020027

**Published:** 2024-02-26

**Authors:** Takuya Omura, Akemi Inami, Taiki Sugimoto, Shuji Kawashima, Takashi Sakurai, Haruhiko Tokuda

**Affiliations:** 1Department of Endocrinology and Metabolism, Hospital, National Center for Geriatrics and Gerontology, 7–430 Morioka-cho, Obu 474-8511, Aichi, Japan; 2Department of Metabolic Research, Research Institute, National Center for Geriatrics and Gerontology, 7–430 Morioka-cho, Obu 474-8511, Aichi, Japan; 3Department of Prevention and Care Science, Research Institute, National Center for Geriatrics and Gerontology, 7–430 Morioka-cho, Obu 474-8511, Aichi, Japan; 4Department of Medicine, University of Washington, Box 359780, 325 Ninth Avenue, Seattle, WA 98104, USA; 5Department of Clinical Laboratory, Hospital, National Center for Geriatrics and Gerontology, 7–430 Morioka-cho, Obu 474-8511, Aichi, Japan

**Keywords:** tirzepatide, time in range, continuous glucose monitoring, elderly

## Abstract

This observational pilot study aimed to investigate continuous glucose monitoring (CGM) metrics in older Japanese patients with type 2 diabetes mellitus (T2DM) using a CGM system (FreeStyle Libre Pro) during the first tirzepatide administration and compare the glycemic control measures before and after the initial injection. The four patients had a mean age of 79.5 years (standard deviation [SD]: 5.8), a mean body mass index of 24.6 kg/m^2^ (SD: 4.7), a mean glycated hemoglobin level of 9.1% (SD: 2.1), and a mean measurement period of 10.5 days (SD: 3.5). After the inclusion of tirzepatide treatment, the mean of time in range, time above range, and time below range changed from 53.2% to 78.9% (*p* = 0.041), 45.8% to 19.7% (*p* = 0.038), and 1.0% to 1.5% (*p* = 0.206), respectively. Improved hyperglycemia reduced the oral hypoglycemic medication in two cases and decreased the frequency of insulin injections in two cases. To elucidate the potential benefits of tirzepatide, future studies should investigate the long-term impact on functional prognosis, safety, and tolerability and distinguish between the use of other weekly agonists, especially in nonobese older Asian patients. However, tirzepatide-associated robust glycemic improvement may simplify diabetes treatment regimens in older patients with T2DM.

## 1. Introduction

Eli Lilly and Company launched tirzepatide (Mounjaro^®^) in the United States on 7 June 2022. Tirzepatide was initially released in Japan on April 18, 2023, including two initiation and maintenance doses (2.5 and 5 mg), followed by four higher doses (7.5, 10, 12.5, and 15 mg), on June 12. Tirzepatide, a new medication based on a natural gastric inhibitory peptide (GIP) sequence, has been structurally modified to bind to glucagon-like peptide 1 (GLP-1) receptors. It is designed based on the amino acid sequence of GIP but is modified to also bind to the GLP-1 receptor, thereby stimulating insulin secretion in a glucose-concentration-dependent manner [1,2]. The insulin-secreting effect of GIP is significantly reduced in patients with chronic hyperglycemic conditions. Additionally, GIP enhances energy accumulation in adipocytes, potentially causing obesity [3]. However, GIP has recently been reported to suppress appetite and induce weight loss effects at concentrations far above physiological levels. A completely new once-weekly injectable formulation has been developed to act on GIP receptors with the accumulation of new results.

The SURPASS J-mono study [4], a domestic phase III clinical trial, was conducted to assess the superiority of weekly tirzepatide doses (5, 10, or 15 mg) over 0.75 mg dulaglutide in 636 Japanese patients with inadequately controlled type 2 diabetes mellitus (T2DM). Participants were randomized to the tirzepatide or dulaglutide groups, and tirzepatide exhibited a significant reduction in glycated hemoglobin (HbA1c) compared with dulaglutide. Adverse events did not significantly differ between the groups. The focus on obese subjects and the minimal representation of older individuals raises questions about generalizability. Additionally, the first-line use of an incretin modulator should be acknowledged.

The body mass index (BMI) of the Japanese population with diabetes is increasing, but it remains significantly lower than that of their Western counterparts. The average BMI of registered patients with T2DM in 2018 was 24.8 kg/m^2^ [5]. The continuous glucose monitoring (CGM) substudy in the SURPASS-3 trial evaluated the association between tirzepatide and hyperglycemic or hypoglycemic time, as well as glycemic variability, compared to insulin degludec in 243 individuals over 52 weeks [6]. The study focused on titration in the SURPASS-3 subpopulation with diabetes (mean BMI: 33.9 kg/m^2^) and revealed that tirzepatide reduced daily glycemic variability and was closely aligned with target ranges. This multicenter, randomized, open-label study predominantly included obese participants, which differs from the Japanese population with T2DM. The limited representation of individuals aged ≥75 years raises uncertainty regarding the applicability of the study to this older population.

The HbA1c is the gold standard for the glycemic control index, but it does not reflect the detailed daily fluctuations in blood glucose levels. The use of CGM provides vital data for “better glycemic management quality” in older populations prone to unconscious hypoglycemia. Furthermore, the simplification of diabetes drug therapy is essential for older patients with impaired cognitive function, visual function, and hand dexterity. Therefore, we investigated the short-term advantages and disadvantages of tirzepatide onset evaluated by CGM in older patients with T2DM in Japan.

## 2. Materials and Methods

### 2.1. Study Design

This observational pilot study investigated short-term changes in glycemic control induced by tirzepatide onset using CGM in Japanese older patients with T2DM.

### 2.2. Participants

Patients with diabetes in our department underwent CGM, specifically when notable blood glucose fluctuations, hypoglycemia unawareness, or both were suspected. Participants signed written informed consent during the FreeStyle Libre Pro sensor attachment.

This study focuses on the first tirzepatide administration in patients with T2DM who actively manage their condition through a combination of dietary control, exercise, and pharmacological treatment, yet continue to suffer insufficient glycemic control. Tirzepatide initiation was well explained and performed. The patients decided when to take weekly tirzepatide injections.

The local ethics committee of the National Centre for Geriatrics and Gerontology approved this exploratory observational study aimed at building evidence on diabetes in older individuals (reception No. 1724).

### 2.3. CGM

CGM was performed using a CGM system (FreeStyle Libre Pro), worn on the upper arm of the nondominant hand for up to 14 days. The data were averaged over the measurement period. Furthermore, CGM was used to measure the appropriate glucose range (70–180 mg/dL, time in range [TIR]), hyperglycemic range (>180 mg/dL, time above range [TAR]), and hypoglycemic range (<70 mg/dL, time below range [TBR]) [7], in addition to sensor glucose and coefficient of variation (%CV).

### 2.4. Category Classification

A joint committee of the Japan Diabetes Society and Japan Geriatrics Society published “Japan Diabetes Society (JDS)/Japan Geriatrics Society (JGS)” to provide safe and effective diabetes care tailored to individual patient conditions [8,9]. Patients were categorized into three classifications based on health status, age, and use of medications that can cause hypoglycemia and comorbidities. Category classification is a comprehensive measure of physical, cognitive, and daily functions in older people with diabetes [10]. The more advanced the category, the worse the patient’s daily function. Patients were classified according to the Daily Functional Scale-8 questionnaire [10].

### 2.5. Statistical Analysis

Mean CGM metrics included normally distributed data and were therefore compared before and after tirzepatide administration using a paired *t*-test. The IBM Statistical Package for the Social Sciences version 28 (Armonk, NY, USA) was used for statistical analyses.

## 3. Case Presentations

(Case 1) An 81-year-old female patient with an 18-year diabetes duration and Category II diabetes reported a history of hypertension, dyslipidemia, hyperuricemia, and osteoporosis. She is currently being treated with glimepiride at 1 mg/day and acarbose at 300 mg/day, with the recent addition of tirzepatide. The glimepiride dose was successfully reduced to 0.5 mg/day after 36 days and was discontinued after 64 days (Table 1).

(Case 2) An 87-year-old male patient with a 13-year diabetes duration and Category I disease reported a history of angina pectoris and polymyalgia rheumatica, for which he takes prednisolone at 3 mg/day. His diabetes is currently being managed with glimepiride at 0.5 mg/day, empagliflozin at 10 mg/day, imeglimine at 2000 mg/day, and dulaglutide at 0.75 mg/week. The dulaglutide supply was limited. Thus, tirzepatide was replaced with dulaglutide. Imeglimine was successfully discontinued after 38 days (Table 2). Furthermore, tirzepatide was also discontinued due to weight loss.

(Case 3) A 74-year-old male patient with an 11-year diabetes duration and Category II cancer reported a preexisting nonfunctioning adrenal tumor, along with arteriosclerosis obliterans, hypertension, and hyperuricemia. His medications included dapagliflozin (5 mg/day), voglibose (0.6 mg/day), pioglitazone (15 mg/day), and insulin degludec/insulin injections (twice a day). Voglibose was discontinued. The frequency of insulin injections was reduced from twice a day to once a day after 22 days (Table 3).

(Case 4) A 76-year-old male patient with approximately a 26-year diabetes duration and Category II reported a history of angina pectoris. The patient received insulin injections four times a day. Insulin aspart injections three times a day were discontinued, and the number of insulin injections was considerably reduced after tirzepatide initiation (Table 4).

Figure 1 and Table 5 describe the course of the four cases.

## 4. Results

The analysis focused on four cases where the initial administration of tirzepatide coincided with CGM measurements conducted for up to 2 weeks. They reported a mean age of 79.5 years (median, 78.5; range, 74–87), a mean BMI of 24.6 kg/m^2^ (median, 23.6; range, 20.4–30.8), a mean glycated hemoglobin level of 9.1% (median, 8.3; range, 7.7–12.1), and a mean measurement period of 10.5 days (standard deviation: 3.5).

A *t*-test was used to compare the CGM indices of the four cases. After the tirzepatide treatment, the mean of TIR, TAR, and TBR improved from 53.2% (standard deviation [SD]: 18.2) to 78.9% (SD: 6.8) (*p* = 0.041), from 45.8% (SD: 18.6) to 19.7% (SD: 7.4) (*p* = 0.038), and from 1.0% (SD: 1.4) to 1.5% (SD: 1.5) (*p* = 0.206), respectively (Figure 2).

## 5. Discussion

Herein, we present four cases of CGM recorded for older Japanese patients with T2DM who received tirzepatide treatment. Immediately after the tirzepatide administration, hyperglycemia improved and TIR increased in older patients with T2DM. Furthermore, the mean TBR increased, but this short-term observation of some patients did not demonstrate significant differences.

The primary aim of this analysis is not to assess the mid-term or monthly effects of tirzepatide treatment but rather to confirm whether there is any occurrence of nocturnal or unintended hypoglycemia after the first injection and whether there are any unexpected adverse events in older patients. The exact timing of glucose lowering after the first administration of tirzepatide is not fully understood. We were able to present new clinical data demonstrating rapid improvement in sensor glucose levels after the first injection of tirzepatide.

Three of the four patients belonged to Category II (mild cognitive impairment to mild dementia or instrumental activity of daily living [ADL] decline and basic ADL independence) [8,9]. The early and potent hypoglycemic effect reduced the number of daily oral hypoglycemic medications and insulin injections. Correcting hyperglycemia while avoiding hypoglycemia is particularly important for older patients with T2DM [11]. Therefore, tirzepatide administration could be a useful treatment option for older patients with T2DM.

Conversely, a strong weight loss effect of tirzepatide has been reported [12,13,14], including rapid weight loss in three of four patients. Correction of obesity may be beneficial in older T2DM patients with obesity, but weight loss may be a risk factor for frailty/sarcopenia. One patient discontinued tirzepatide because of weight loss. An increase in the tirzepatide dose from the initial dose is expected to improve the weight loss effect. Therefore, longer-term observation within the Japanese patient population, which includes a large proportion of older and nonobese patients, is essential.

During short-term follow-up, eGFR values worsened in two cases (Cases 1 and 4) and improved in two cases (Cases 2 and 3). In the results of the post hoc analysis of renal outcomes in the SURPASS-4 trial, eGFR values appeared to decrease initially, suggesting an initial dip [15]. The clinical significance of this variation in eGFR values was unclear. However, clinicians should be aware of the eGFR value, which may decrease in the early stages of administration.

The present analysis is not an interventional study, but rather an examination of a small number of cases; the observation period for CGM was short. The duration of CGM monitoring before and after tirzepatide administration varied from case to case. Although there was little difference in the daily routines of the patients in this study between weekdays and weekends, the potential influence of life-cycle effects on the CGM indicator cannot be ignored. Furthermore, the small sample size raises concerns about the generalizability of the statistics derived from the statistical analysis to the older population as a whole. However, the experience gained from its application with older patients was deemed valuable.

Although it is important to conduct future interventional and long-term observational studies, it is also crucial to establish the safe use of tirzepatide. Additionally, while the powerful effect of various agents in reducing blood glucose levels and body weight is remarkable, it can also lead to adverse effects such as hypoglycemia, weight loss, frailty, and sarcopenia, especially in older patients. Therefore, when using GLP-1 receptor agonists, it is important to clarify the optimal formulation for various older individuals.

## 6. Conclusions

The CGM index immediately improved after tirzepatide administration. In the short term, tirzepatide may potentially decrease the requirement for multiple medications and streamline treatment regimens. The long-term effects of tirzepatide in a population that includes older and nonobese individuals will be investigated in future studies.

## Figures and Tables

**Figure 1 geriatrics-09-00027-f001:**
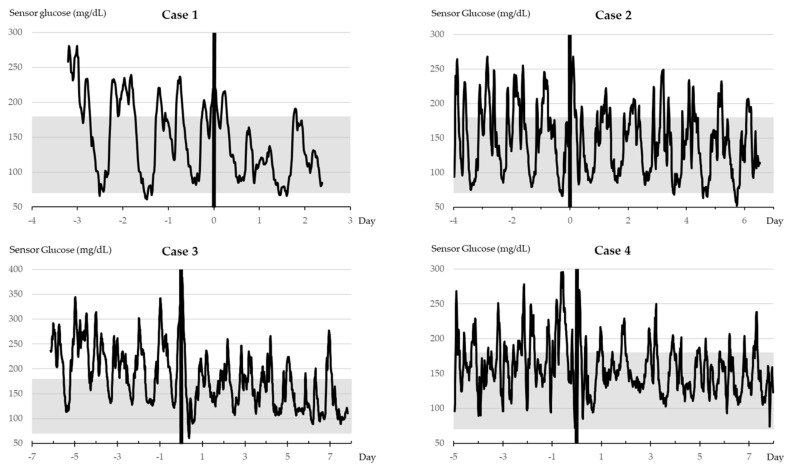
Tirzepatide initiation and CGM. Vertical bold lines indicate the timing of tirzepatide administration. The grayscale denotes the TIR range.

**Figure 2 geriatrics-09-00027-f002:**
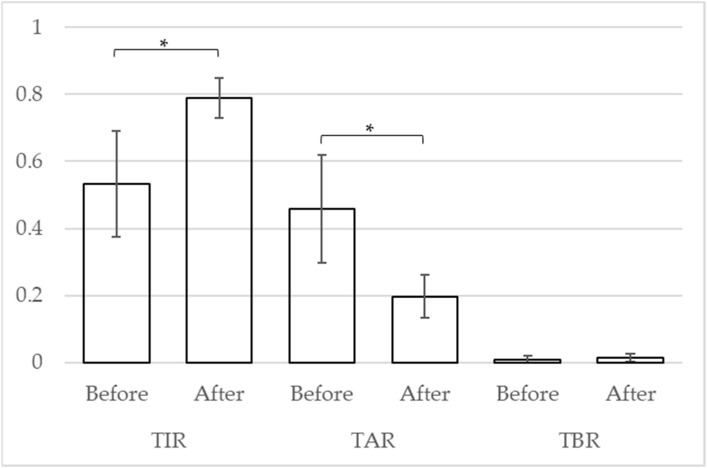
Change in CGM indicators. The vertical axis indicates the proportion of each indicator. * *p* < 0.05.

**Table 1 geriatrics-09-00027-t001:** Treatment course of Case 1.

	7 Days before the First Injection	23 Days after the First Injection	36 Days after the First Injection
HbA1c (%)	12.1	-	9.8
LDL/HDL/TG (mg/dL)	84/52/171	78/53/147	94/68/126
eGFRcr (mL/min/1.73 m^2^)	40.2	34.9	34.2
AST/ALT (IU/L)	38/44	24/44	25/46
Body weight (kg)	75.6	73.9	-

LDL cholesterol, LDL; HDL cholesterol, HDL; triglyceride, TG; aspartate aminotransferase, AST; alanine aminotransferase, ALT.

**Table 2 geriatrics-09-00027-t002:** Treatment course of Case 2.

	4 Days before the First Injection	23 Days after the First Injection	38 Days after the First Injection
HbA1c (%)	7.8	-	7.3
LDL/HDL/TG (mg/dL)	82/51/99	-	63/52/103
eGFRcr (mL/min/1.73 m^2^)	70.7	-	73.8
AST/ALT (IU/L)	17/10	-	16/9
Body weight (kg)	54.0	52.7	-

LDL cholesterol, LDL; HDL cholesterol, HDL; triglyceride, TG; aspartate aminotransferase, AST; alanine aminotransferase, ALT.

**Table 3 geriatrics-09-00027-t003:** Treatment course of Case 3.

	6 Days before the First Injection	22 Days after the First Injection
HbA1c (%)	8.8	7.8
LDL/HDL/TG (mg/dL)	98/64/150	73/50/101
eGFRcr (mL/min/1.73 m^2^)	50.8	54.6
AST/ALT (IU/L)	16/11	15/8
Body weight (kg)	71.2	71.0

LDL cholesterol, LDL; HDL cholesterol, HDL; triglyceride, TG; aspartate aminotransferase, AST; alanine aminotransferase, ALT.

**Table 4 geriatrics-09-00027-t004:** Treatment course of Case 4.

	5 Days before the First Injection	15 Days after the First Injection	21 Days after the First Injection
HbA1c (%)	7.7	7.5	-
LDL/HDL/TG (mg/dL)	89/56/71	85/46/69	-
eGFRcr (mL/min/1.73 m^2^)	70.5	63.6	-
AST/ALT (IU/L)	21/21	18/18	-
Body weight (kg)	53.0	-	53.8

LDL cholesterol, LDL; HDL cholesterol, HDL; triglyceride, TG; aspartate aminotransferase, AST; alanine aminotransferase, ALT.

**Table 5 geriatrics-09-00027-t005:** Changes in CGM indices before and after tirzepatide administration.

	Observation Periods	Before			After		
	(Days)	TIR	TAR	TBR	TIR	TAR	TBR
Case 1	7	52.6%	44.5%	2.9%	85.2%	11.8%	3.1%
Case 2	8	67.5%	31.4%	1.0%	76.6%	21.0%	2.4%
Case 3	14	27.6%	72.4%	0.0%	70.4%	29.2%	0.4%
Case 4	13	65.1%	34.9%	0.0%	83.4%	16.6%	0.0%

## Data Availability

Anonymized data will be available on request to any qualified investigator after approval by the Ethics Committee.

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
