# Peer review of "Tirzepatide and Glycemic Control Metrics Using Continuous Glucose Monitoring in Older Patients with Type 2 Diabetes Mellitus: An Observational Pilot Study"

_geriatrics, 2024, doi:10.3390/geriatrics9020027_

Round 1

Reviewer 1 Report

Comments and Suggestions for Authors

Case reports usually deal with single individual patient (description of the patient and associated problem, physical examination and test results, treatment and actual outcome).

This article describes four elderly patients with type 2 diabetes to whom  tirzepatide was administered (treatment) - their glucose was continuously measured for a period (several days), and glycemic control measures before and after the treatment were compared. In addition to the before-after comparison, some other data are listed that are relevant for this type of problem.

The idea is for the article to be the starting point for further research related to the problem of tirzepatide therapy in elderly patients, which the authors state in the conclusion.

The article also has a results section in which the authors observe those 4 patients as a sample for which they estimate the average age and standard deviation. This is not necessary because nothing is done with a group of four, but with each patient individually. However, if the authors want to emphasize the age of the patients, then a description with median and range is more adequate. Furthermore, differences in continuously measured glucose before-after, measured for the TIR, TAR, TBR range, are statistically tested for each patient. Given that it is about percentages, it is more appropriate to apply non-parametric testing (not t-test).

Tables 1-4: titles (before x days, after y days) should be described in more details. E.g. „before x days“ – does it mean x days before the terapy was administered? If it is so, what is meaning of „after y days“?

Author Response

Dear reviewer,

We would like to express our sincere appreciation for considering our revised Case Report entitled " Tripeptide and Glycemic Control Metrics Using Continuous Glucose Monitoring in Older Patients with Type 2 Diabetes Mellitus: An Observational Pilot Study (geriatrics-2814988)." Thanks to the reviewer's valuable comments, we have significantly improved our manuscript.

We look forward to hearing from you at your earliest convenience.

Sincerely,

Takuya OMURA

Reviewer 2 Report

Comments and Suggestions for Authors

The manuscript presents the results of a pilot study investigating the effects of GLP-1 Tirzepatide on glycemic control in older Japanese patients with type 2 diabetes.

The objective is of interest to the community, being the first study conduced on such a population. However, the results are only supported by 4 subjects data (including 1 discontinued), and not statistically significant amount of CGM data. These limitations should be further stressed in the discussion and conclusion.
In addition, the study design and method lack of relevant information, affecting the manuscript interpretability. 
For these reasons, I suggest major review should be performed before publication.

Find below more specific comments.

1. Abstract. Specify the total CGM monitoring duration (as mean and sd), as it characterizes the interpretation of the results

2. Abstract. Specify that participants were under additional medications

3. Introduction, Line 33. The name of the drug is wrongly spelled, it is Mounjaro.

4. Introduction, Line 61. The name of the compound is wrongly spelled, it is degludec.

5. Methods, Line 91. The sentence is not completed, should be rephrased.

6. Methods, Line 95. The sentence about informed consent should be moved in the previous section "Participants".

7. Methods. Study Design should be more specific. Refer to Figure 1 to support the description of the study design. For how long were participants supposed to wear CGM? When was Tirzepatide administered? How many in-clinic visits? What are inclusion/exclusion criteria? What was the overall quality of glucose control at baseline? Were there any other assessments (e.g., Frailty classification)

8. Methods. Please clarify the rationale for the category classification. How has this been used in the study? What was the expectation? How can this impact a bigger study?

9. Results. I do not understand why some follow-up results are not available for some participants (e.g., HbA1c for case 1, all but body weight for case 2, etc.)

10. Results. While the recommended CGM duration was 2 weeks, the actual mean and sd of wear time should be reported.

11. Results. Is there a reason why eGFR increases in case 3, while it decreases in case 1 and case 4?

12. Discussion. Clarify that 2 weeks of CGM is not enough to reliably assess differences. In the literature >14 days of CGM are recommended to reliably evaluate glucose control (I recommend citing https://doi.org/10.1111/dom.14483), so 14 days before treatment plus 14 days after treatment would have been the most appropriate solution. Please clarify what drove such a shorter monitoring duration? In addition, discuss how the weekday/weekend variability might impact the results, given that participants might be more exposed to hypoglycemia in the weekend, as compared to weekdays.

13. Discussion. I suggest adding a paragraph about "Lesson learnt" from this pilot study. How can this experience can be leveraged for a wider clinical trial? (e.g., effect size estimation, study duration, attrition, etc.)

14. Discussion. I suggest adding a paragraph about "Study limitations" of this pilot study. Short monitoring duration can be affected by participants' lifestyle (see previous point 9); low sample size leads to low power, and potentially unrealiable pvalues; 

15. Discussion. Why the mild dementia of 3 participants have been stressed? What is the relationship with the study outcomes?

16. Discussion. How was frailty index computed? Fried Frailty Score? Please clarify in the methods.

17. Discussion. Was there a pre-specified condition for discontinuation? Why one participant discontinued because of weight loss? Please, clarify in the study design.

Comments on the Quality of English Language

Quality of English is acceptable.

Author Response

(The authors gave the same response as above.)

Round 2

Reviewer 2 Report

Comments and Suggestions for Authors

Thank you for addressing all comments. The paper is ready for publication.